# Sexual behaviours and sexual health outcomes among young adults with limiting disabilities: findings from third British National Survey of Sexual Attitudes and Lifestyles (Natsal-3)

Elizabeth Holdsworth,[1] Viktoriya Trifonova,[2] Clare Tanton,[2] Hannah Kuper,[1] Jessica Datta,[1] Wendy Macdowall,[1] Catherine H Mercer[2]

[1]Public Health and Policy, London School of Hygiene and Tropical Medicine, London, UK
[2]Infection and Population Health, University College London, London, UK

**Correspondence to**
Elizabeth Holdsworth;
elizabeth.holdsworth@lshtm.ac.uk

## ABSTRACT

**Objective** To explore whether the sexual behaviours and sexual health outcomes of young adults with self-reported disabilities that they perceive limit their activities ('limiting disability') differ from those without disability.

**Design** Complex survey analyses of cross-sectional probability sample survey data collected between September 2010 and August 2012 using computer-assisted personal interviewing and computer-assisted self-interview.

**Setting** British general population.

**Participants** 7435 women and men aged 17–34 years, resident in private households in Britain, interviewed for the third National Survey of Sexual Attitudes and Lifestyles.

**Main outcome measures** Self-reported sexual behaviour and sexual health outcomes.

**Results** Approximately 1 in 10 participants reported having a limiting disability. Sexual behaviours were similar between those with limiting disability and those without, with a few exceptions. Women and men with limiting disability were less likely to report having sexual partner(s) (past year, adjusted ORs (AORs) for age and social class: AORs: 0.71, 0.75, respectively). Women with limiting disability were more likely to report having same-sex partner(s) in the past 5 years (AOR: 2.39). Differences were seen in sexual health outcomes, especially among women; those with limiting disability were more likely to report having experienced non-volitional sex (ever, AOR: 3.08), STI diagnoses (ever, AOR: 1.43) and sought help/advice regarding their sex life (past year, AOR: 1.56). Women with limiting disability were also more likely to feel distressed/worried about their sex life than those without limiting disability (AORs: 1.61). None of these associations were seen in men.

**Conclusions** Young adults with limiting disability, especially women, are more likely to report adverse sexual health outcomes than those without, despite comparatively few behavioural differences. It is important to ensure that people with disabilities are included in sexual health promotion and service planning, and targeted policy and programme interventions are needed to address negative sexual health outcomes disproportionally experienced by people with disabilities.

### Strengths and limitations of this study

► This paper presents the results of the analysis of a large-scale, nationally representative survey, which achieved a response rate in line with other major social surveys completed in Britain around the same time.
► It is one of few quantitative studies to explore whether sexual behaviour and sexual health outcomes differ between people with limiting disability and those with no disability and the only one we know of to date in Britain.
► A strength of third National Survey of Sexual Attitudes and Lifestyles is that it used computer-assisted personal interview and specifically computer-assisted self-interview to minimise reporting bias for more sensitive questions.
► As a cross-sectional survey, chronology cannot always be determined and nor can causality in the associations we show be inferred; for example, we have no information about the duration of disability, and whether a participant's disability preceded their first heterosexual intercourse.

## INTRODUCTION

The United Nations Convention on the Rights of Persons with Disabilities defines disability as 'those who have long-term physical, mental, intellectual or sensory impairments which in interaction with various barriers may hinder their full and effective participation in society on an equal basis with others'.[1] It is estimated that there are 1 billion people living with a disability worldwide,[2] and in Britain, there are over 11 million people with a limiting long-term illness, impairment or disability, equating to almost one in six of the population.[3] From both human rights and public health perspectives, it is important that sexual and reproductive health services are inclusive of this large group, since sexual health and

sexual satisfaction are recognised as significant predictors of quality of life and general life satisfaction.[4 5] However, it is argued that the sexuality and sexual health of people with disabilities have traditionally been neglected.[6 7] This may be a result of misconceptions that disabled people are asexual[7–11] or because the sexual well-being of people with disabilities is of less concern than rehabilitation and other health priorities.[12 13] This is despite evidence from qualitative research highlighting the same need for sexual health services among those with disabilities as in the wider population.[5] Negative experiences with healthcare professionals are commonly reported by people with disabilities; these include a failure to discuss sex because professionals do not think the topic pertinent.[9] Findings also identify unmet need for support for problems with sexual function[14 15] and sexual satisfaction.[5] However, there is an absence of reliable, empirical evidence from large-scale, population-level surveys that explore the sexual lifestyles and experiences of disabled people in Britain.

Britain's third National Survey of Sexual Attitudes and Lifestyles (Natsal-3), a probability sample survey, offers an opportunity to address these evidence gaps. Earlier analyses of Natsal-3 data highlighted differences in sexual experiences between people with disabilities and those without including the increased prevalence of 'non-volitional' or 'non-consensual' sex reported by people with disabilities,[16] and the association between poor health and decreased sexual activity and satisfaction.[17] This paper seeks to explore in greater depth the sexual behaviours and sexual health outcomes reported by people with and without limiting disabilities, specifically among young people as the age group at the highest risk of negative sexual health outcomes.[18–20]

## METHODS
### Participants and procedures
Natsal-3 was a stratified probability sample survey of 15 162 men and women aged 16–74 years, resident in households in Britain, who were interviewed in 2010–2012. Details of the methodology are described in detail elsewhere,[21] and the questionnaire and technical report are available online (www.natsal.ac.uk). Participants provided oral consent. Participants completed the survey through a combination of face-to-face computer-assisted personal interview (CAPI) and computer-assisted self-interview (CASI) for the more sensitive questions.

In the CAPI section of the interview, all participants were asked '*Do you have any long-standing illness, disability or infirmity?*' in which '*long-standing*' was defined as '*anything that has troubled you over a period of time, or that is likely to affect you over a period of time*'. Participants who answered 'yes' were routed to the question: '*Does this limit your activities in any way?*'. Participants who reported 'yes' were defined for the purposes of this analysis as having 'limiting disability'. This definition concurs with that used for the Equality Act in the UK[22] and the United Nations Convention on the Rights of Persons with Disabilities.[1] In this paper, we compared those reporting limiting disability with those reporting no long-standing illness or disability. This means that our comparative analyses exclude participants reporting a non-limiting disability, because they cannot easily be categorised either as 'disabled' or 'non-disabled' according to the prevailing conceptualisation of disability.[23]

To obtain information about self-reported clinical diagnoses of a range of health conditions, interviewers in the CAPI showed participants cards listing a number of different conditions and asked whether they had been diagnosed with any of those listed. These included mental and physical health conditions (eg, depression, arthritis, cardiac diseases, diabetes, epilepsy, broken hip or pelvis, backache or bone or muscle disease) lasting for more than 3 months in the past year.

Participants were also asked about their first sexual experiences in the CAPI through showcards, and then in the CASI they were asked questions about their experience of sexual practices, numbers of sexual partners in different timeframes, their recent partnerships, sexual function and sexual health, including sexually transmitted infection (STI) diagnosis. The interview concluded with another CAPI, which included standard demographic questions about educational attainment, employment, sexual identity and ethnicity.

The overall estimated response rate to Natsal-3 was 57.7%, while among those aged 16–34 years, it was estimated as 64.8%.[24] For this analysis, we focused on participants aged 17–34 years, excluding 16 year olds, as one of our key demographic variables is educational attainment and therefore all participants in our sample will have had the chance to attain qualifications obtained by the UK school leaving age of 16 years. We can also differentiate between those who left school at that point and those who went on to study for qualifications typically gained aged 17+ years.

### Statistical analysis
We completed statistical analyses using the survey functions of Stata (V.14.1) to take account of the stratification, weighting and clustering of the Natsal-3 dataset. The data were weighted to adjust for the unequal probabilities of selection and non-response and corrected for differences in gender, age and regional distribution according to the UK 2011 census, so that the data are broadly representative of the resident general population in Britain.[24]

We initially estimated the prevalence of reporting a limiting disability among all young people, and also the prevalence of reporting a disability that was *not* perceived as limiting. We then examined the prevalence of health conditions that were asked about in Natsal-3 according to whether participants reported a limiting disability or no disability at all, in order to provide context, although we recognise that these conditions may or may not be the cause of participants' limiting disability (online supplementary table 1). Our binary variable of reporting

'limiting disability' or 'no disability' was then initially treated as a dependent (outcome) variable to examine how prevalence varies by key sociodemographic factors. In subsequent analyses, we used this variable as an independent (response) variable to consider how reporting sexual behaviours and sexual health outcomes vary for those with a limiting disability in comparison with those without. We present prevalence estimates and adjusted ORs with 95% CIs. We used multivariable logistic regression to calculate ORs adjusted for potential confounding variables, specifically age, and individual-level socioeconomic status (measured according to the National Statistics Socio-economic Classification[25]).

### Role of funding source
This research paper received no specific grant from any funding agency in the public, commercial or not-for-profit sectors. The Natsal-3 study was supported by grants from the Medical Research Council (G0701757) and the Wellcome Trust (084840),with contributions from the Economic and Social Research Council and Department of Health. The sponsors of the original Natsal-3 study had no role in study design, data collection, data analysis, data interpretation or writing of this paper.

### Patient and public involvement
Patients or members of the public were not involved in the development, design or conduct of this study.

### RESULTS
### Prevalence of limiting disability and most commonly reported health conditions
Of all participants aged 17–34 years, 11.0% (95% CI 10.0% to 12.1%) of women and 8.2% (95% CI 7.1% to 9.4%) of men reported having an illness, disability or infirmity that limited their activities (table 1).

A further 9.9% (95% CI 8.9% to 10.9%) of all women and 8.1% (95% CI 7.1% to 9.3%) of all men in this age group reported a disability that they did not perceive as limiting their activities (data not shown). These participants with non-limiting disability (439 women and 255 men) correspond to approximately half of all participants in this age range who reported a disability and are excluded from subsequent analyses. Overall, the majority of women and men with limiting disability reported having one or more physical and/or mental health condition (76.5% women; 71.8% men; conditions shown in online supplementary table 1). Relative to those reporting no disability, mental health conditions were reported by a large proportion of those with limiting disability: 50% of women (AOR 5.19) and 45% of men (AOR 6.25). Depression was the most commonly reported mental health condition by men and women with limiting disability. Physical health conditions were also more frequently reported by those with limiting disability, with 50% of men (AOR 12.67) and 52% of women (AOR 10.26) reporting one or more physical

health condition. Having difficulty or being unable to walk up a flight of stairs and having backache or bone or muscle disease for more than 3 months in the past year were the physical conditions most commonly reported by participants with limiting disability. Those with limiting disability had high levels of comorbidity with 40.6% of women and 39.9% of men with limiting disability reporting two or more physical and/or mental health conditions (AOR 19.2 and AOR 42.3, respectively).

### Variation in the reporting of limiting disability by key sociodemographic characteristics
Prevalence of limiting disability increased with age in men, but not women (table 1). Among women, prevalence of limiting disability was lower in those of black/black British ethnicity than those of other ethnicities and higher among those not currently in a steady relationship. There was no overall statistically significant association with relationship status for either gender. Although the numbers of participants not identifying as heterosexual was small, prevalence of limiting disability was higher among women who did not, including after adjustment relative to women identifying as heterosexual. There was an association with socioeconomic status for both genders, with those reporting currently having no job (AOR 5.88 for men and 2.94 for women) or being a student (AOR 1.78 for men and AOR 1.47 for women) more likely to report limiting disability. Men and women with no academic qualifications were also more likely to report having limiting disability. We found no variation by deprivation area of residence as measured by the Index of Multiple Deprivation.[26]

### Association between limiting disability and sexual behaviour
Those with limiting disability were no different to those without limiting disability in terms of the number of sexual partners reported (including those where condoms were not used), or in the frequency of sex reported (table 2). In terms of reporting sexual practices, vaginal sex in the past month was the only practice where there was a difference, with this less commonly reported by women with limiting disability than those without (AOR 0.75). Compared with women with no limiting disability, those with limiting disability were more likely to report having same-sex partner(s) in the last 5 years (AOR 2.39), but this was not observed in men. Differences were also observed in terms of where male and female participants met their most recent partner. For example, those reporting limiting disability were more likely to have done so via the internet than those with no disability (9.5% vs 4.7% for women and 10.9% vs 5.4% for men). Women with limiting disability reported a shorter time between meeting and first sex with their most recent partner than women with no disability and were more likely to report having just, or recently, met their most recent partner when they first had sex together (AOR 1.49 for within 24 hours). These associations were not observed in men.

**Table 1** Variation in the reporting of limiting disability by key sociodemographic characteristics among Natsal-3 participants aged 17–34, by gender

| | Women | | | Men | | |
|---|---|---|---|---|---|---|
| | Unweighted, weighted denominators | % (95% CI) reporting limiting disability | AOR* (95% CI) of reporting limiting disability | Unweighted, weighted denominators | % (95% CI) reporting limiting disability | AOR* (95% CI) of reporting limiting disability |
| All | 3953/2228 | 11.0 (10.0 to 12.1) | | 2786/2284 | 8.2 (7.1 to 9.4) | |
| Age (year) | | | P=0.2486 | | | P<0.0001 |
| 17–19 | 694/369 | 10.4 (8.0 to 13.4) | Reference category | 598/378 | 5.1 (3.5 to 7.4) | Reference category |
| 20–24 | 1054/632 | 10.2 (8.4 to 12.4) | 0.99 (0.68 to 1.46) | 805/648 | 6.5 (4.8 to 8.8) | 1.5 (0.87 to 2.59) |
| 25–29 | 1264/626 | 11.2 (9.5 to 13.1) | 1.2 (0.82 to 1.76) | 806/652 | 9.4 (7.5 to 11.6) | 2.8 (1.68 to 4.66) |
| 30–34 | 941/600 | 12.0 (10.0 to 14.3) | 1.33 (0.89 to 1.99) | 577/606 | 10.5 (7.9 to 13.9) | 3.15 (1.81 to 5.48) |
| Ethnicity | | | P=0.0039 | | | P=0.3076 |
| White | 3327/1818 | 11.6 (10.5 to 12.8) | Reference category | 2350/1851 | 8.7 (7.6 to 10.0) | Reference category |
| Mixed | 130/70 | 16.1 (9.2 to 26.8) | 1.41 (0.72 to 2.76) | 81/68 | 7.9 (3.6 to 16.6) | 0.78 (0.33 to 1.86) |
| Asian/Asian British | 259/184 | 7.9 (5.2 to 11.8) | 0.52 (0.31 to 0.86) | 199/224 | 5.7 (2.0 to 15.1) | 0.61 (0.20 to 1.85) |
| Black/black British | 156/100 | 3.8 (1.5 to 9.2) | 0.29 (0.11 to 0.73) | 99/89 | 5.3 (2.1 to 12.9) | 0.53 (0.20 to 1.42) |
| Chinese/other† | 71/50 | 7.0 (2.0 to 21.5) | 0.51 (0.14 to 1.86) | 51/48 | 3.0 (0.6 to 13.0) | 0.28 (0.06 to 1.25) |
| Relationship status | | | P=0.0568 | | | P=0.6962 |
| Living with a partner | 1738/1083 | 10.5 (9.1 to 12.1) | Reference category | 983/956 | 8.7 (6.8 to 10.9) | Reference category |
| In a steady relationship, not cohabiting | 915/447 | 10.3 (8.4 to 12.5) | 1.13 (0.83 to 1.53) | 616/453 | 7.7 (5.8 to 10.3) | 1.22 (0.77 to 1.92) |
| No steady relationship | 1253/673 | 12.5 (10.6 to 14.7) | 1.39 (1.06 to 1.83) | 1148/848 | 7.6 (6.2 to 9.4) | 1.14 (0.76 to 1.70) |
| Sexual identity‡ | | | P<0.0001 | | | P=0.1344 |
| Heterosexual/straight | 3779/2127 | 10.3 (9.4 to 11.4) | Reference category | 2678/2201 | 8.1 (7.0 to 9.3) | Reference category |
| Gay/lesbian | 43/24 | 18.7 (9.4 to 33.6) | 2.21 (1.02 to 4.77) | 61/45 | 13.4 (7.0 to 23.9) | 1.99 (1.00 to 3.99) |
| Bisexual | 105/59 | 28.5 (18.8 to 40.8) | 3.72 (2.12 to 6.51) | 29/22 | 8.9 (2.6 to 26.5) | 1.37 (0.40 to 4.76) |
| National Statistics Socio-economic Classification | | | P<0.0001 | | | P<0.0001 |
| Manager/professional | 940/565 | 8.3 (6.6 to 10.4) | Reference category | 698/639 | 6.1 (4.1 to 9.1) | Reference category |
| Intermediate | 705/393 | 10.2 (8.1 to 12.8) | 1.29 (0.90 to 1.86) | 353/299 | 11.6 (8.5 to 15.7) | 2.12 (1.22 to 3.70) |
| Semi to routine/routine | 1221/632 | 12.6 (10.8 to 14.8) | 1.73 (1.26 to 2.37) | 983/762 | 8.6 (6.9 to 10.6) | 1.76 (1.08 to 2.88) |

Continued

**Table 1** Continued

| | Women | | | Men | | |
|---|---|---|---|---|---|---|
| | Unweighted, weighted denominators | % (95% CI) reporting limiting disability | AOR* (95% CI) of reporting limiting disability | Unweighted, weighted denominators | % (95% CI) reporting limiting disability | AOR* (95% CI) of reporting limiting disability |
| No job currently | 313/173 | 19.5 (15.1 to 24.8) | 2.94 (1.96 to 4.42) | 115/83 | 19.0 (12.5 to 27.8) | 5.88 (3.00 to 11.53) |
| Student | 750/453 | 9.5 (7.3 to 12.3) | 1.47 (0.96 to 2.25) | 624/4487 | 6.1 (4.2 to 8.7) | 1.78 (0.96 to 3.29) |
| Academic qualifications§ | | | P=0.0002 | | | P<0.0001 |
| No academic qualifications | 341/167 | 19.9 (15.4 to 25.3) | Reference category | 218/171 | 20.4 (15.7 to 26.2) | Reference category |
| Qualifications typically gained at age 16 years | 1203/605 | 14.3 (12.3 to 16.6) | 0.79 (0.55 to 1.13) | 853/657 | 9.7 (7.8 to 11.9) | 0.47 (0.31 to 0.71) |
| Studying for or have attained further academic qualifications | 2253/1355 | 8.7 (7.5 to 10.0) | 0.48 (0.33 to 0.72) | 1625/1364 | 6.2 (4.8 to 7.9) | 0.29 (0.18 to 0.48) |
| Quintile of Index of Multiple Deprivation (IMD) | | | P=0.4923 | | | P=0.6785 |
| Least deprived (1 and 2) | 1238/702 | 11.0 (9.2 to 13.2) | Reference category | 962/756 | 7.0 (5.4 to 8.9) | Reference category |
| 3 | 785/462 | 10.5 (8.5 to 12.9) | 0.88 (0.64 to 1.21) | 529/431 | 7.7 (4.9 to 11.8) | 1.04 (0.60 to 1.83) |
| Most deprived (4 and 5) | 1930/1064 | 11.1 (9.7 to 12.7) | 0.85 (0.66 to 1.11) | 1295/1097 | 9.2 (7.6 to 11.0) | 1.16 (0.82 to 1.63) |

*OR adjusted for age and social class.
†Chinese and other subcategories were merged because of small numbers in these categories.
‡Excludes those reporting 'other' to the question about sexual identity as this was reported by too few participants to provide robust estimates.
§Applies only to respondents aged 17+ years.

**Table 2** Variations in the reporting of key sexual behaviours among Natsal-3 participants aged 17–34 years by limiting disability status and gender

| | Women | | | Men | | |
|---|---|---|---|---|---|---|
| | % (95% CI) of those reporting no disability (n=3495/1983) | % (95% CI) of those reporting limiting disability (n=458/245) | P values | % (95% CI) of those reporting no disability (n=2539/2098) | % (95% CI) of those reporting limiting disability (n=247/186) | P values |
| Number of partners*, past year | | | | | | |
| 0 | 11.8 (10.5 to 13.2) | 14.6 (11.3 to 18.7) | | 12.7 (11.3 to 14.3) | 14.4 (10.2 to 19.9) | |
| 1 | 68.4 (66.5 to 70.2) | 63.0 (57.8 to 67.8) | | 59.5 (57.2 to 61.7) | 60.0 (52.7 to 66.8) | |
| ≥2 | 19.8 (18.4 to 21.4) | 22.4 (18.2 to 27.2) | | 27.8 (25.9 to 29.8) | 25.6 (20.1 to 32.0) | |
| AOR†‡ (0 vs ≥1) | 1 | 0.71 (0.50 to 1.02) | 0.061 | 1 | 0.75 (0.48 to 1.19) | 0.226 |
| Number of partners* without a condom, past year | | | | | | |
| 0 | 25.4 (23.7 to 27.2) | 27.4 (22.9 to 32.5) | | 31.3 (29.1 to 33.5) | 28.9 (22.9 to 35.7) | |
| 1 | 64.0 (62.1 to 65.9) | 61.2 (56.1 to 66.0) | | 55.6 (53.2 to 57.9) | 57.1 (49.7 to 64.2) | |
| ≥2 | 10.6 (9.5 to 11.8) | 11.4 (8.8 to 14.7) | | 13.2 (11.7 to 14.7) | 14.0 (10.0 to 19.3) | |
| AOR†‡ | 1 | 1.13 (0.82 to 1.55) | 0.45 | 1 | 1.22 (0.81 to 1.85) | 0.34 |
| Number of occasions of sex*, past 4 weeks | | | | | | |
| 0–2 | 45.4 (43.4 to 47.5) | 50.1 (44.8 to 55.4) | | 47.4 (45.2 to 49.7) | 51.5 (43.8 to 59.1) | |
| 3–4 | 17.2 (15.8 to 18.8) | 19.0 (15.1 to 23.7) | | 17.1 (15.5 to 18.9) | 12.9 (8.6 to 18.8) | |
| 5+ | 37.3 (35.4 to 39.3) | 30.8 (26.1 to 36.0) | | 35.4 (33.3 to 37.6) | 35.6 (28.9 to 42.9) | |
| AOR†‡ | 1 | 1.24 (0.98 to 1.55) | 0.07 | 1 | 1.24 (0.87 to 1.75) | 0.232 |
| Vaginal sex, past month | 70.3 (68.6 to 72.0) | 65.4 (60.2 to 70.3) | | 66.5 (64.4 to 68.5) | 67.0 (60.4 to 73.0) | |
| AOR†‡ | 1 | 0.75 (0.59 to 0.95) | 0.016 | 1 | 0.93 (0.68 to 1.27) | 0.636 |
| Given/received oral sex*, past month | 54.2 (52.3 to 56.2) | 50.61 (45.56 to 55.64) | | 56.0 (53.8 to 58.2) | 55.4 (48.2 to 62.3) | |
| AOR†‡ | 1 | 0.88 (0.71 to 1.09) | 0.252 | 1 | 1.01 (0.75 to 1.37) | 0.923 |
| Genital contact without intercourse*, last month | 53.7 (51.8 to 55.6) | 50.9 (45.8 to 56.0) | | 53.8 (51.5 to 56.1) | 47.6 (40.4 to 55.0) | |
| AOR†‡ | 1 | 0.91 (0.73 to 1.14) | 0.409 | 1 | 0.82 (0.59 to 1.14) | 0.238 |
| Same-sex partner(s), past 5 years | 5.0 (4.2 to 5.8) | 10.4 (7.5 to 14.2) | | 3.2 (2.5 to 4.0) | 4.1 (2.4 to 7.0) | |
| AOR†‡ | 1 | 2.39 (1.61 to 3.54) | <0.0001 | 1 | 1.35 (0.73 to 2.48) | 0.339 |
| Where first met most recent partner | | | | | | |
| School/work | 36.0 (34.1 to 37.9) | 29.0 (24.4 to 34.0) | | 41.2 (38.9 to 43.5) | 28.0 (20.6 to 36.7) | |
| Online/internet dating | 4.7 (4.0 to 5.6) | 9.5 (6.9 to 13.0) | 0.0007 | 5.4 (4.4 to 6.7) | 10.9 (7.1 to 16.4) | |
| Always known each other/ neighbour | 7.0 (6.0 to 8.1) | 8.1 (5.7 to 11.4) | | 4.9 (4.0 to 6.0) | 6.0 (3.5 to 10.2) | |
| Public place | 20.3 (18.8 to 21.9) | 18.9 (15.1 to 23.4) | | 21.8 (20.0 to 23.7) | 18.2 (13.3 to 24.5) | |
| Other | 32.0 (30.2 to 33.8) | 34.5 (29.6 to 39.8) | | 26.6 (24.6 to 28.8) | 36.9 (30.0 to 44.4) | 0.0003 |
| Time between first meeting most recent partner and first sex | | | | | | |
| 24 hours or less | 5.2 (4.4 to 6.2) | 9.9 (7.1 to 13.7) | | 9.2 (7.9 to 10.6) | 11.4 (7.6 to 16.7) | |

**Table 2** Continued

| | Women | | | Men | | |
|---|---|---|---|---|---|---|
| | % (95% CI) of those reporting no disability (n=3495/1983) | % (95% CI) of those reporting limiting disability (n=458/245) | P values | % (95% CI) of those reporting no disability (n=2539/2098) | % (95% CI) of those reporting limiting disability (n=247/186) | P values |
| Between 1 day and 1 week | 7.7 (6.7 to 8.9) | 9.8 (7.1 to 13.4) | | 9.8 (8.5 to 11.2) | 8.9 (5.6 to 13.8) | |
| Between 1 week and 6 months | 56.2 (54.2 to 58.2) | 50.9 (45.3 to 56.4) | | 54.5 (52.1 to 56.9) | 49.9 (41.7 to 58.1) | |
| Between 6 months and 5 years | 26.0 (24.3 to 27.7) | 23.3 (18.9 to 28.4) | | 22.6 (20.6 to 24.8) | 23.9 (16.5 to 33.2) | |
| 5 years or more | 4.9 (4.0 to 5.9) | 6.0 (4.0 to 8.9) | | 3.9 (3.1 to 5.0) | 6.0 (3.3 to 10.5) | |
| AOR†‡ | 1 | 1.49 (1.09 to 2.02) | 0.012 | 1 | 1.01 (0.70 to 1.47) | 0.94 |
| Condom not used on first occasion with most recent partner§ | 35.4 (33.4 to 37.5) | 40.4 (35.0 to 46.0) | | 38.2 (35.8 to 40.6) | 46.9 (38.0 to 56.1) | |
| AOR †‡ | 1 | 1.12 (0.88 to 1.43) | 0.343 | 1 | 1.24 (0.84 to 1.83) | 0.275 |
| Relationship status at first sex with most recent partner | | | | | | |
| Just met/had met recently | 20.8 (19.1 to 22.5) | 33.8 (28.8 to 39.2) | | 29.4 (27.1 to 31.7) | 32.3 (25.1 to 40.5) | |
| Know each other/used to be in a relationship | 25.1 (23.3 to 26.9) | 21.8 (17.6 to 26.7) | | 27.9 (25.7 to 30.1) | 25.2 (18.9 to 32.7) | |
| Steady relationship/living together/married | 54.2 (52.1 to 56.2) | 44.4 (38.9 to 50.0) | | 42.8 (40.3 to 45.3) | 42.5 (33.6 to 51.9) | |
| AOR†‡ | 1 | 1.93 (1.48 to 2.51) | P<0.0001 | 1 | 1.14 (0.78 to 1.66) | 0.493 |

*Opposite sex and/or same-sex partner.
†OR adjusted for age and social class.
‡Adjusted OR for reporting the responses in bold font (for those variables with ≥2 response options) relative to 'no disability'.
§Respondents who only had oral sex on the most recent occasion were excluded.
AOR, adjusted OR; Natsal-3, third National Survey of Sexual Attitudes and Lifestyles.

### Circumstances of sexual debut by disability status

We found differences by limiting disability status in the circumstances of sexual debut among women (table 3). Women with limiting disability were more likely to report earlier sexual debut (aged under 16 years at first heterosexual intercourse versus aged 16 years or older, AOR 1.64) and to report that they had to be persuaded or were forced (AOR 1.94) at first sex. Women with limiting disability were also more likely to be categorised as lacking 'sexual competence'[i] at first heterosexual intercourse (AOR 1.31 relative to those reporting no disability).

### Variations in the reporting of sexual health outcomes by disability status

Women with limiting disability were more likely to report having ever experienced non-volitional sex than women without disability (AOR 3.08), with a higher AOR also for attempted non-volitional sex (AOR 2.50) (table 4). Women with limiting disability were also more likely to STI diagnosis/es (ever) (AOR 1.52) and relatedly having attended a sexual health clinic (ever, AOR 1.26). Women with limiting disability were more likely than those without disability to disclose that they were distressed or worried about their sex lives (AORs 1.61),

[i] On the assumption that first intercourse should, ideally, be characterised by absence of duress and regret, autonomy of decision, and use of a reliable method of contraception, four variables relating to circumstances: regret, willingness, autonomy, and contraception at first

intercourse, were used as criteria in the construction of a measure of sexual competence.[47]

**Table 3** Variations in the reporting of circumstances relating to sexual debut among Natsal-3 participants aged 17–34 years by limiting disability status and gender

| | Women | | | Men | | |
|---|---|---|---|---|---|---|
| | % (95% CI) of those reporting no disability (n=3495/1983) | % (95% CI) of those reporting limiting disability (n=458/245) | P values | Men % (95% CI) of those reporting no disability (n=2539/2098) | % (95% CI) of those reporting limiting disability (n=247/186) | P values |
| **Age at first heterosexual intercourse (year)** | | | | | | |
| **13–15** | 27.8 (26.1 to 29.6) | 39.6 (34.6 to 44.9) | | 29.8 (27.8 to 32.0) | 36.4 (29.4 to 44.0) | |
| 16–17 | 43.3 (41.3 to 45.3) | 39.8 (34.7 to 45.0) | | 39.6 (37.3 to 41.9) | 35.8 (28.7 to 43.6) | |
| 18–19 | 16.4 (15.0 to 18.0) | 12.3 (9.2 to 16.3) | | 20.0 (18.2 to 22.0) | 19.6 (14.3 to 26.3) | |
| ≥20 | 12.5 (10.9 to 14.1) | 8.3 (5.8 to 11.7) | | 10.6 (9.1 to 12.2) | 8.2 (3.2 to 19.3) | |
| AOR*† | 1 | 1.64 (1.29 to 2.09) | 0.0001 | 1 | 1.36 (0.98 to 1.89) | 0.0682 |
| **Willingness at first heterosexual intercourse‡** | | | | | | |
| Both willing | 82.8 (81.2 to 84.3) | 76.0 (71.1 to 80.4) | | 91.1 (89.6 to 92.3) | 88.9 (83.5 to 92.7) | |
| Respondent more willing | 1.3 (0.9 to 1.8) | 2.4 (1.1 to 5.1) | | 3.5 (2.6 to 4.6) | 2.1 (0.9 to 5.0) | |
| Partner more willing, respondent also willing | 6.7 (5.7 to 7.9) | 4.5 (2.8 to 7.4) | | 3.4 (2.7 to 4.3) | 7.1 (4.1 to 12.1) | |
| Respondent had to be persuaded | 8.1 (7.1 to 9.3) | 13.3 (10.0 to 17.5) | | 1.9 (1.4 to 2.7) | 1.9 (0.7 to 5.2) | |
| Respondent was forced | 1.1 (0.8 to 1.6) | 3.7 (2.1 to 6.3) | | 0.1 (0.0 to 0.4) | 0 | |
| AOR*† | 1 | 1.94 (1.41 to 2.66) | <0.0001 | – | – | |
| Lack of sexual competence at first heterosexual intercourse | 48.8 (46.9 to 50.7) | 57.8 (52.6 to 62.8) | | 44.8 (42.4 to 47.3) | 47.4 (39.5 to 55.4) | |
| AOR*† | 1 | 1.31 (1.04 to 1.65) | 0.0218 | 1 | 0.95 (0.68 to 1.33) | 0.7788 |
| Lack of autonomy at first heterosexual intercourse§ | 39.1 (37.1 to 41.1) | 36.4 (31.3 to 41.8) | | 47.6 (45.3 to 50.0) | 43.4 (35.7 to 51.5) | |
| AOR*† | 1 | 0.90 (0.70 to 1.14) | 0.376 | 1 | 0.82 (0.58 to 1.16) | 0.266 |
| **Opinion now of timing of first heterosexual intercourse¶** | | | | | | |
| Should have waited longer | 32.3 (30.5 to 34.2) | 38.2 (33.3 to 43.4) | | 16.6 (15.0 to 18.4) | 22.4 (17.0 to 28.9) | |
| Should not have waited so long | 3.1 (2.4 to 3.9) | 4.9 (2.9 to 8.1) | | 7.0 (5.9 to 8.3) | 5.1 (2.7 to 9.3) | |
| About the right time | 64.6 (62.8 to 66.4) | 56.9 (51.6 to 62.0) | | 76.3 (74.3 to 78.3) | 72.5 (65.5 to 78.6) | |
| AOR*† | 1 | 1.21 (0.96 to 1.52) | 0.1136 | 1 | 1.38 (0.95 to 2.00) | 0.0871 |
| Reliable contraception not used at first sex¶ | 14.0 (12.7 to 15.4) | 18.7 (14.9 to 23.3) | | 17.8 (15.9 to 19.7) | 24.4 (18.5 to 31.5) | |
| AOR*† | 1 | 1.16 (0.85 to 1.59) | 0.335 | 1 | 1.21 (0.84 to 1.75) | 0.3134 |

*OR adjusted for age and social class.
†Adjusted OR for reporting the responses in bold font (for those variables with ≥2 response options) relative to 'no disability'.
‡Not sufficient numbers to report OR for men.
§Reasons for first intercourse: peers doing it; bit drunk; smoked some cannabis; taken some other drugs.
¶Applies to respondents not forced.
AOR, adjusted OR; Natsal-3, third National Survey of Sexual Attitudes and Lifestyles.

and one-third of women with limiting disability reported having sought help or advice for their sex life in the past year and were more likely to have done so than women with no disability (approximately one-quarter; AOR 1.56). None of these associations were observed among men.

**Table 4** Variations in the reporting of key sexual health outcomes among Natsal-3 participants aged 17–34 years by limiting disability status and gender

| Sexual health outcome | Women | | | | Men | | | |
|---|---|---|---|---|---|---|---|---|
| | % (95% CI) of those reporting no disability (n=3495/1983) | % (95% CI) of those reporting limiting disability (n=458/245) | AOR* (95% CI) for reporting outcome if reported limiting disability | P values | % (95% CI) of those reporting no disability (n=2539/2098) | % (95% CI) of those reporting limiting disability (n=247/186) | AOR* 95% (CI) for reporting outcome if reported limiting disability | P values |
| Experienced non-volitional sex | 6.9 (6.0 to 8.0) | 19.5 (15.7 to 24.0) | 3.08 (2.28 to 4.16) | <0.0001 | 1.3 (0.9 to 1.9) | 2.1 (0.9 to 4.6) | 1.57 (0.64 to 3.90) | 0.322 |
| Experienced attempted non-volitional sex | 16.0 (14.7 to 17.4) | 33.0 (28.3 to 38.1) | 2.50 (1.96 to 3.19) | <0.0001 | 4.2 (3.4 to 5.2) | 5.7 (3.3 to 9.7) | 1.38 (0.73 to 2.60) | 0.321 |
| Ever diagnosed with a STI† | 37.8 (35.9 to 39.6) | 47.2 (42.1 to 52.4) | 1.52 (1.21 to 1.91) | 0.0003 | 13.0 (11.5 to 14.6) | 12.4 (8.6 to 17.4) | 0.89 (0.57 to 1.38) | 0.598 |
| Ever attended a sexual health (GUM) clinic‡ | 42.1 (40.1 to 44.1) | 46.3 (41.2 to 51.6) | 1.26 (1.01 to 1.58) | 0.044 | 35.0 (32.8 to 37.2) | 35.7 (28.9 to 43.0) | 1.04 (0.75 to 1.45) | 0.811 |
| Ever had a pregnancy that ended in an abortion | 12.2 (11.0 to 13.4) | 15.3 (12.0 to 19.1) | 1.25 (0.92 to 1.68) | 0.148 | | | | |
| Distressed/worried about sex life | 10.9 (9.7 to 12.1) | 16.7 (13.1 to 21.2) | 1.61 (1.18 to 2.21) | 0.003 | 9.7 (8.5 to 11.0) | 14.6 (10.3 to 20.2) | 1.48 (0.97 to 2.26) | 0.068 |
| Sought help/advice for sex life, past year | 25.2 (23.6 to 26.9) | 33.8 (28.9 to 39.1) | 1.56 (1.22 to 2.00) | 0.0004 | 19.6 (17.9 to 21.4) | 18.5 (13.7 to 24.7) | 1.01 (0.68 to 1.48) | 0.977 |

*OR adjusted for age and social class.

## DISCUSSION

This paper presents the results of the analysis of a large-scale, nationally representative survey, in which we explored whether sexual behaviour and sexual health outcomes differ between people with and without limiting disability. It is one of few quantitative studies to do so, and the only one we know of to date in Britain. Disability that limited activities affected around 1 in 10 people in this relatively young age group (17–34 years). Around three-quarters of respondents with a limiting disability reported having one or more physical and/or mental health conditions. The main finding from these analyses is that, while young adults with disabilities in Britain report broadly similar sexual behaviour to young adults without disabilities, they are more likely to experience adverse sexual health outcomes. This is especially so for women. Of note, women with limiting disability were significantly more likely to have experienced sex against their will, STI diagnosis/es, an earlier sexual debut and lack 'sexual competence' at first sex, including less frequent use of reliable contraception. While we did not find these associations for men, both women and men with limiting disability were more likely to report greater distress and less satisfaction with their sex lives than their peers.

There are relatively few comparable studies available and none reporting on a British population. In the USA, the Minnesota Adolescent Health study found few differences in sexual behaviours among young people with and without chronic physical conditions but, like our study, found poorer outcomes among those with chronic conditions including a higher proportion who had a history of sexual abuse and STI diagnosis.[27] The US National Longitudinal Study of Adolescent Health found that physically disabled young people were as likely to be sexually active as their peers, but that young women with physical disabilities were more vulnerable to non-consensual sex.[28] Our findings support existing evidence that women with disabilities are a group at higher risk of experiencing non-volitional sex,[16] sexual assault[29–31] and intimate partner violence.[30–32]

Our finding that people with limiting disability experience more distress and less satisfaction with their sex lives may be due to people with severe physical illnesses experiencing sexual difficulties as a direct result of their condition.[14] Other studies, including qualitative research, have reported higher levels of dissatisfaction or distress about sex life among people with disabilities that suggests that people with physical disability have the same sexual needs and desires as people without disability, but that their body image, sexual self-esteem, sexual satisfaction and life satisfaction may be lower.[5 33]

In women with limiting disability, we also observed a shorter time between meeting and first sex with their most recent partner than in women with no limiting disability. Previous research on stereotypes associated with disability and sexuality suggests that a woman who feels sexually disenfranchised or who has lower sexual esteem as a result of her disability may be more likely to have sex with a partner with whom she is less emotionally invested.[34–36] However, having sex with someone soon after meeting may not, in itself, be a negative outcome if the experience is mutually desired, safe, pleasurable, free of coercion, discrimination and violence.[4] Nonetheless, this may not always be the case given the higher prevalence of adverse sexual health outcomes for young adults with limiting disability observed in the Natsal-3.

There are limitations that need to be taken into consideration when interpreting the results from our study. Natsal-3 achieved a response rate of 57.7% overall in line with other major social surveys completed in Britain around the same time,[37 38] although the response rate was higher among young people, this paper's study population.[24] Non-response weighting was used such that the data broadly reflect the distribution of key demographic variables according to census data; however, selection bias is a potential issue. In this respect, it is important to acknowledge that Natsal-3's sampling frame meant that only people resident in private households in Britain were sampled, excluding people living in institutions who may be more likely to have limiting disabilities. In addition, despite Natsal-3's large sample size (including oversampling people in our study's age range), a relatively small proportion of participants were of non-white British ethnicity reflecting Britain's ethnic composition.[39] Unlike Natsal-2,[40] Natsal-3 did not oversample ethnic minorities, therefore limiting the power to detect ethnic differences as reflected in some wide CIs and requiring us to use broad categories of self-reported ethnicity (eg, black/black British) in which there exists great heterogeneity.

A strength of Natsal-3 is that it used CAPI and specifically CASI to minimise reporting bias for the more sensitive questions. Nonetheless, the data are self-reported, which are subject to recall and social desirability bias. Furthermore, as a cross-sectional survey, chronology cannot always be determined and nor can causality in the associations we show be inferred. We have no information about the duration of disability, and whether, for example, a participant's disability preceded their first heterosexual intercourse. We restricted our analysis to focus on people with limiting disability in line with national and international legislation and policy[1 22] and so we have not included those who considered their disability as non-limiting. While those with non-limiting disability could be explored in a future analysis, it is worth noting that earlier analyses of Natsal-3 considered the associations between general health status and measures of sexual behaviour and sexual well-being.[17]

The study included people who considered themselves to have a limiting disability rather than focusing specifically on people with particular impairment types, for example, sensory impairment, as is the case in most previous studies.[9 14 41] However, there is a lack of information on the nature and severity of the impairment underlying the disability, which could help us further elucidate the relationship between disability and sexual

health. In an attempt to provide context, we presented data on a number of health conditions and considered how this varied according to whether participants perceived themselves to have a limiting disability. Both mental and physical health conditions were more commonly reported by people with limiting disability than those without, supporting our use of this measure of disability. However, it was not possible to determine whether a participant's limiting disability was as a result, even in part, of the conditions reported, or whether these conditions were experienced in addition to their limiting disability.

Our findings have important implications for policy and practice. First, limiting disability was common in this relatively young age group and, for the most part, sexual behaviour of people with disabilities was similar to that among those without disability. This points to the need for young people with limiting disabilities to be represented and included in sexual health promotion alongside their contemporaries. Second, that some negative outcomes are more commonly reported by this group suggest that targeted efforts are also needed, which may need to be newly developed as they are currently lacking. Of note, non-volitional sex, which may need targeted policy and programme interventions. Sexual assault is frequently unreported to the police or authorities, and research has shown that reporting is even less likely among people with a disability.[42] When a report is made, support following sexual assault neither targets the circumstances of, nor meets the needs of, people with disability.[42–45] Interventions for distress about sex lives may also require targeted policy and programme interventions. These should include awareness raising and/or educational interventions for health professionals, as evidence suggests a reluctance or failure to discuss sex with individuals with disabilities as it is not seen as pertinent[9] or aspects of the clinical, institutional and broader social environments may undermine their ability to promote sexual health.[46] The study findings and recommendations will be of interest to disabled people's organisations and sexual health advocates, as well as policy makers and health professionals. There are also implications for further research, including the need for qualitative research to understand the relationship between experiencing disability, distress and satisfaction about sex.

**Contributors** EH conceived the study. EH, CHM, CT, HK, JD and WM contributed to the design of the study. VT and CT conducted analysis of the data, and all authors contributed to the interpretation of the data. EH drafted the article, and it was critically revised for important intellectual content by EH, CHM, CT, HK, JD and WM. All authors contributed to the final approval of the version to be published. All authors had full access to all the data (including statistical reports and tables) in the study and can take responsibility for the integrity of the data and the accuracy of the data analysis.

**Funding** The Natsal grant for the original data collection is funded by Wellcome and MRC.

**Competing interests** None declared.

**Patient consent** Not required.

**Ethics approval** Natsal-3 was granted ethical approval from the Oxford A NHS Research Ethics Committee (reference: 09/H0604/27).

**Provenance and peer review** Not commissioned; externally peer reviewed.

**Data sharing statement** The Natsal-3 dataset has been archived at the UK Data Archive at the University of Essex and is accessible by academic researchers.

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
