## [Reviewer comments · BMJ Open]

ARTICLE DETAILS

TITLE (PROVISIONAL)	Sexual behaviours and sexual health outcomes among young adults with limiting disabilities: findings from third British National Survey of Sexual Attitudes and Lifestyles (Natsal-3).
AUTHORS	Holdsworth, Elizabeth; Trifonova, Viktoriya; Tanton, Clare; Kuper, Hannah; Datta, Jessica; Macdowall, Wendy; Mercer, Catherine

VERSION 1 – REVIEW

REVIEWER	Tom Shakespeare Norwich Medical School, UEA, UK
REVIEW RETURNED	11-Sep-2017

GENERAL COMMENTS	This is an excellent paper, on an important topic. I find it very interesting and reassuring that the quantitative data reinforces the qualitative research which is familiar to me. The finding that around 10% of young adults have disability is about right, as is the socioeconomic gradient and gender disparity. Again, the finding that disabled people are having sex, but in fewer numbers than non-disabled is sad but expected. So is the higher risk of gender-based violence, as found in literature, for example the special issue of Journal of Interpersonal Violence on disability. Nor is the finding that disabled people are more likely to have depression: the twin experiences of adverse health conditions and prejudice/discrimination in society produces that, presumably. I am sceptical that this explains poorer sexual experiences. My suspicion is that the majority of disabled people with poor mental health are not taking anti-depressants, so be careful with this claim. It may be that because of lack of partner choice, people are settling for partners with whom they are unhappy, or staying in unhappy relationships. It may be that people have physical pains or restrictions which make sex more difficult, or lack energy. There are lots of reasons why sex may not be as positive for some disabled people as we would hope. The conclusions regarding sex education are important, and we could also suggest that disabled people's organisations and sexual health advocates should pay attention to this data and recommendations. Really timely and interesting, well worth publishing with or without minor tweaks.
--

REVIEWER	Gillian Eastgate Queensland Centre for Intellectual and Developmental Disability, University of Queensland, Australia
REVIEW RETURNED	17-Nov-2017

GENERAL COMMENTS	Some references are somewhat old. There are newer studies in this area that should be looked at and probably included There is no information regarding ethics approval. The reader is
---

	directed to the report on the main NATSAL-3 study for details of the overall study design and methodology. My understanding is that the ethical process would have been detailed in this report but this is not explicitly stated. If this report is to stand alone then the reader also needs to know where to find details of the ethics process (perhaps just add this in the reference to the main study) Inability to determine chronology and causality are noted as limitations of the study. I agree, but I think that a further (and very major) limitation is that the definition of disability is extremely broad and encompasses conditions that may be better defined as chronic illness than as disability. This can't be changed in a supplementary report on an existing dataset but I think it needs to be acknowledged Otherwise, this is a well-written report in an area where there are large gaps in knowledge, and as such merits publication
--	---

REVIEWER	zohreh Shahhosseini Mazandaran University of Medical Sciences, Sari, Iran
REVIEW RETURNED	26-Nov-2017

GENERAL COMMENTS	Thanks a lot for your invitation and also your comprehensive and valuable research article. From my point of view, the manuscript "Sexual behaviors and sexual health outcomes among young adults with limiting disabilities: findings from third British National Survey of Sexual Attitudes and Lifestyles" is written with high accuracy and it requires only few changes: Manuscript need briefly english editing is some sections Abstract word count was approximately high so it is proposed to write summarized. Some references need editing. Type of disabilities is not specified in the text of manuscript exactly. Please check the keywords based on the MESH - The reviewer provided a marked copy with additional comments. Please contact the publisher for full details.
---

REVIEWER	Megan Quinn East Tennessee State University, USA
REVIEW RETURNED	23-Dec-2017

GENERAL COMMENTS	The article is well written with a clear research question, appropriate and well explained methodology, and interesting and clearly defined results. Further, the discussion and conclusions provide additional explanation and comparison of the results and public health and policy implications.
---

REVIEWER	Mark Lyons-Amos University of Portsmouth, United Kingdom
REVIEW RETURNED	23-Jan-2018

GENERAL COMMENTS	Reviewer's report for Sexual behaviours and sexual health outcomes among young adults with limiting disabilities: findings from third British National Survey of Sexual Attitudes and Lifestyle (Natsal-3) This analysis is an observational and descriptive report on various sexual health outcomes, with a focus on young adults with self reported limiting long term illness. The focus on the paper is primarily establishing the prevalence of sexual health outcomes, and
---

	as such uses a relatively simple methodology comprised of descriptive tables and regression analysis. The paper makes a distinct contribution: the sexual outcomes of the group under analysis are underexplored as established by the authors, and are certainly a vulnerable group worthy of policy attention. I feel that with some refinement, this paper could be well sited within the literature to make a good descriptive paper. I include a list of major and minor revisions below Major comments 1. A stronger justification of why those with a non-limiting disability are not examined in detail. The authors claim that this is due to an ambiguous definition, but thereafter seem to make reference to this group reasonably distinctly. Is this group really not identifiable from the data? It seems to me that including this group would be a natural mid point between the two groups currently analysed in terms of establishing directional effect of disability. 2. The tables presented highlight a number of potential correlated (ethnicity etc.) which are potentially important confounders, but these are not included in the regression models. Why is this? It would be highly useful to see how these variables interplay when identifying particularly risky groups. 3. The current regression models include only age and educational attainment as a proxy for socioeconomic status. This is extremely problematic, not least because education attainment will be heavily affected by limiting longterm illness. The authors appear to have more standard SES measures in the dataset, why not use these instead? Minor comments pp. 7. Ln 27 The preferred style is Stata, rather than STATA
--	---

REVIEWER	Emmanuel F Drabo Stanford University
REVIEW RETURNED	19-Feb-2018

GENERAL COMMENTS	This study helps fill a critical gap in knowledge about the sexual behaviors and sexual health outcomes of individuals with disabilities in Britain. However, the study has some important analytical limitations that need to be addressed in order to convincingly support its conclusions. Addressing some of the concerns that I raise below would help enhance the study and its contributions. Minor comments: A clear working definition of "disability" needs to be stated very early in the introduction. ORs/AORs are difficult to interpret from a practical standpoint, hence all AOR estimates should be translated into marginal effects, which are more intuitive. On pp. 11, line 7 there appears to be a typo: "one so" should be "done so" I suppose.
---

	In the main text (e.g. lines 49-53, pp.8), the authors provide the prevalence of the health outcomes for those reporting a limiting disability and provide the AORs. This is not sufficient information; they should also provide the prevalence of the health conditions for those reporting no disability. Major comments: Table 1. (pp. 20): consistency needed in the presentation of the unweighted/weighted denominators values in the cells (use "/" or ", " in all cells; my preference is "/" to avoid confusions with the "thousands" notation). A description of the full sample (women + men) should be included in the table, or at the very least, characterize how different the two subgroups are, so that we can better interpret the estimates. We would also want to better understand the drivers of the gender differences in outcomes between the disables and non-disabled. The authors should also present the marginal effects, and use those in the main text, because they are more intuitive and have a natural interpretation than AORs. The authors do not present any information regarding the significance of the differences in estimates for women and men. Are these gender differences significant and meaningful? What statistical tests (if any) are used to test these differences? Line 37, pp.20: more clarification needed on the age and education adjustments; what specific model(s) is(are) being estimated? Are the authors interacting age and education with other covariates? Table 2. pp.22: Does this model adjust for other covariates in Table 1? If not, why? For example, given the significant racial/ethnic differences in disability status (Table 1), one would want to also control for this variable.
--	--

VERSION 1 – AUTHOR RESPONSE

Reviewer(s)' Comments to Author:

Reviewer: 1

Reviewer Name: Tom Shakespeare

Institution and Country: Norwich Medical School, UEA, UK

Please state any competing interests: I have written extensively on disability and sexuality. I have worked with Hannah Kuper in an advisory capacity.

Please leave your comments for the authors below:

This is an excellent paper, on an important topic. I find it very interesting and reassuring that the quantitative data reinforces the qualitative research which is familiar to me. The finding that around 10% of young adults have disability is about right, as is the socioeconomic gradient and gender disparity. Again, the finding that disabled people are having sex, but in fewer numbers than non-disabled is sad but expected. So is the higher risk of gender-based violence, as found in literature, for example the special issue of Journal of Interpersonal Violence on disability. Nor is the finding that disabled people are more likely to have depression: the twin experiences of adverse health conditions and prejudice/discrimination in society produces that, presumably.

I am sceptical that this explains poorer sexual experiences. My suspicion is that the majority of disabled people with poor mental health are not taking anti-depressants, so be careful with this claim. It may be that because of lack of partner choice, people are settling for partners with whom they are unhappy, or staying in unhappy relationships. It may be that people have physical pains or restrictions which make sex more difficult, or lack energy. There are lots of reasons why sex may not be as positive for some disabled people as we would hope.

Authors' response: We thank the reviewer for their supportive comments. We have edited the paper to remove the line relating to anti-depressants and lack of sexual satisfaction (p.12).

The conclusions regarding sex education are important, and we could also suggest that disabled people's organisations and sexual health advocates so should pay attention to this data and recommendations. Really timely and interesting, well worth publishing with or without minor tweaks.

Authors' response: As suggested, we have added a concluding line about the findings and recommendations being of interest to disabled people's organisations and sexual health advocates (p.15)

Reviewer: 2

Reviewer Name: Gillian Eastgate

Institution and Country: Queensland Centre for Intellectual and Developmental Disability, University of Queensland, Australia

Please state any competing interests: None declared

Please leave your comments for the authors below

Some references are somewhat old. There are newer studies in this area that should be looked at and probably included

Authors' response: We have referenced some newer studies (e.g. reference numbers 7, 11, 45, 46) There is no information regarding ethics approval. The reader is directed to the report on the main NATSAL-3 study for details of the overall study design and methodology. My understanding is that the ethical process would have been detailed in this report but this is not explicitly stated. If this report is to stand alone then the reader also needs to know where to find details of the ethics process (perhaps just add this in the reference to the main study)

Authors' response: We apologise for omitting this information, and we have now added in details of Natsal-3's ethical approval to the first paragraph of the Methods (p.6).

Inability to determine chronology and causality are noted as limitations of the study. I agree, but I think that a further (and very major) limitation is that the definition of disability is extremely broad and encompasses conditions that may be better defined as chronic illness than as disability. This can't be changed in a supplementary report on an existing dataset but I think it needs to be acknowledged

Authors' response: As explained in the second paragraph of the Methods (p.6), we defined disability in this study as participants of Natsal-3 who defined themselves as having "long-standing illness, disability or infirmity" Specifically, all participants were asked "Do you have any long-standing illness, disability or infirmity?" in which "long-standing" was defined as "anything that has troubled you over a period of time, or that is likely to affect you over a period of time". Participants who answered "yes" were routed to the question: "Does this limit your activities in any way?" and participants who reported "yes" were defined for the purposes of this analysis as having "limiting disability". This definition of disability (i.e. not chronic disease alone) is as used under the Equality Act in the UK and the United Nations Convention on the Rights of Persons with Disabilities, and complies with the International Classification of Functioning, Disability and Health (ICF) conceptualisation of disability, as originally cited. We now begin our paper with referring to the UN Convention document, and cite these three references to the Methods to support our study's definition of disability.

Otherwise, this is a well-written report in an area where there are large gaps in knowledge, and as such merits publication

Authors' response: We thank the Reviewer for their concluding comments in support of our paper.

Reviewer: 3

Reviewer Name: Zohreh Shahhosseini

Institution and Country: Mazandaran University of Medical Sciences, Sari, Iran

Please state any competing interests: None declared

Please leave your comments for the authors below

Thanks a lot for your invitation and also your comprehensive and valuable research article. From my point of view, the manuscript "Sexual behaviors and sexual health outcomes among young adults with limiting disabilities: findings from third British National Survey of Sexual Attitudes and Lifestyles" is written with high accuracy and it requires only few changes:

Manuscript need briefly english editing in some sections

Abstract word count was approximately high so it is proposed to write summarized.

Some references need editing.

Please check the keywords based on the MESH

Authors' response: We have checked and edited the paper sections, references, abstract, and keywords.

Type of disabilities is not specified in the text of manuscript exactly.

Authors' response: We are not clear what the Reviewer means by this point. In the Discussion section, we have noted as a study limitations that there is a lack of information on the nature and severity of the impairment underlying the disability or chronic illness, and that unfortunately we cannot determine whether a participant's limiting disability was as a result of the conditions reported, or whether these conditions were experienced in addition to their limiting disability. In an attempt to provide context, as a Supplementary Table we present data on a number of health conditions and considered how this varied according to whether or not participants perceived themselves to have a limiting disability, long-term illness, or infirmity. We trust this detail addresses the Reviewer's concern.

Reviewer: 4

Reviewer Name: Megan Quinn

Institution and Country: East Tennessee State University, USA

Please state any competing interests: None declared

Please leave your comments for the authors below

The article is well written with a clear research question, appropriate and well explained methodology, and interesting and clearly defined results. Further, the discussion and conclusions provide additional explanation and comparison of the results and public health and policy implications.

Authors' response: We thank the reviewer for their supportive comments and are pleased that they had no issues with our paper.

Reviewer: 5

Reviewer Name: Mark Lyons-Amos

Institution and Country: University of Portsmouth, United Kingdom

Please state any competing interests: None declared

Please leave your comments for the authors below

Comments in attached document

This analysis is an observational and descriptive report on various sexual health outcomes, with a focus on young adults with self reported limiting long term illness. The focus on the paper is primarily

establishing the prevalence of sexual health outcomes, and as such uses a relatively simple methodology comprised of descriptive tables and regression analysis. The paper makes a distinct contribution: the sexual outcomes of the group under analysis are underexplored as established by the authors, and are certainly a vulnerable group worthy of policy attention.

Authors' response: We are grateful of the Reviewer's favourable comments.

I feel that with some refinement, this paper could be well sited within the literature to make a good descriptive paper. I include a list of major and minor revisions below

Major comments

1. A stronger justification of why those with a non-limiting disability are not examined in detail. The authors claim that this is due to an ambiguous definition, but thereafter seem to make reference to this group reasonably distinctly. Is this group really not identifiable from the data? It seems to me that including this group would be a natural mid point between the two groups currently analysed in terms of establishing directional effect of disability.

Authors' response: By definition, a disability is an impairment that limits activities or participation, and we have used this definition for the purposes of our paper. As noted in response to Reviewer 2's comments, our definition is in line with that used in the UK's Equality Act and the United Nations Convention on the Rights of Persons with Disabilities, and complies with the prevailing ICF conceptualisation of disability. As noted above, we now begin our paper with referring to the UN Convention document, and cite these references in the Methods as further justification of our study's definition of disability.

As we now note in our Limitations, while non-limiting disability could be the focus of a future analysis, it is worth noting that previous analyses of the Natsal-3 data have looked at the associations between general health and wellbeing more broadly and sexual health outcomes (see Field et al, Lancet 2013, cited as reference #14). In contrast, this paper sets out to look specifically at how limiting disability is associated with sexual behaviour and sexual health outcomes.

2. The tables presented highlight a number of potential correlated (ethnicity etc.) which are potentially important confounders, but these are not included in the regression models. Why is this? It would be highly useful to see how these variables interplay when identifying particularly risky groups.

Authors' response: We adjusted for key sociodemographic characteristics in our multivariable models where multicollinearity was not likely to be an issue (e.g. we did not adjust for educational attainment and SES). We did not adjust for ethnicity as (a) we did not do this as we observed no association between ethnicity and disability for men, and (b) the numbers of participants from 'non-white' ethnic groups were considered too small to meaningfully adjust for this sociodemographic characteristic – a limitation we now flag in our Discussion (see p.13). In addition, the bivariate analyses allow the interested reader to identify particularly risky groups, possibly better than having presented a parsimonious model which may 'explain away' associated factors, which themselves might be helpful indicators of risk in public health practice.

3. The current regression models include only age and educational attainment as a proxy for socioeconomic status. This is extremely problematic, not least because education attainment will be heavily affected by limiting longterm illness. The authors appear to have more standard SES measures in the dataset, why not use these instead?

Authors' response: Measuring young people's socioeconomic circumstances is challenging as concluded from a systematic review by Sheringham et al (STI 2013; doi: 10.1136/sextrans-2011-050223). We agree that educational attainment is far from ideal as a marker of SES, especially for young people as many will not have completed their education at the time of interview. For this reason, our educational attainment had the following three categories: [1] no academic qualifications; [2] qualifications typically gained at age 16; [3] studying for/have attained further academic qualifications). Furthermore, and as explained in the Methods, these categories are applicable to everyone in our study sample as the lower age limit was 17y (rather than 16y as for the parent population, Natsal-3) reflecting how the school leaving age, at the time of fieldwork was 16y.

The Reviewer suggested using SES instead of educational attainment, but this too is problematic as the SES variable available in Natsal-3 - NS-SEC - is based on Standard Occupational Classification, which means that SES is not all that meaningful as many young people are yet to establish their careers and/or still in full-time education. Nevertheless, we re-ran the analyses and now adjust for NS-SEC instead of education, although there is little difference in the AORs and so our findings remain unchanged. We have updated the Methods section accordingly and also include a reference to the NS-SEC as our measure of SES.

Minor comments

pp. 7. Ln 27 The preferred style is Stata, rather than STATA

Authors' response: We thank the Reviewer for alerting us to this error which we have now corrected.

Reviewer: 6

Reviewer Name: Emmanuel F Drabo

Institution and Country: Stanford University

Please state any competing interests: None declared

Please leave your comments for the authors below

This study helps fill a critical gap in knowledge about the sexual behaviors and sexual health outcomes of individuals with disabilities in Britain.

Authors' response: We thank the Reviewer for recognising how our paper fills an evidence-gap.

However, the study has some important analytical limitations that need to be addressed in order to convincingly support its conclusions. Addressing some of the concerns that I raise below would help enhance the study and its contributions.

Minor comments:

A clear working definition of "disability" needs to be stated very early in the introduction.

Authors' response: We thank the reviewer for their comments. Our working definition is given in the Objective statement of our Abstract, and we have added in the United Nations Convention on the Rights of Persons with Disabilities definition of disability into the introduction.

On pp. 11, line 7 there appears to be a typo: "one so" should be "done so" I suppose.

Authors' response: We thank the reviewer for alerting us to the typo, it is now corrected.

ORs/AORs are difficult to interpret from a practical standpoint, hence all AOR estimates should be translated into marginal effects, which are more intuitive.

Also: The authors should also present the marginal effects, and use those in the main text, because they are more intuitive and have a natural interpretation that AORs.

Authors' response: There are many different ways of quantifying associations. In this paper, we use odds ratios as in most other Natsal papers to date, including two series in the Lancet (e.g. www.thelancet.com/themed/natsal). In addition, we present ORs alongside prevalence estimates, the

corresponding 95% confidence intervals, and denominators, and as such enable the reader to compare prevalences and not just the relative risk.

In the main text (e.g. lines 49-53, pp.8), the authors provide the prevalence of the health outcomes for those reporting a limiting disability and provide the AORs. This is not sufficient information; they should also provide the prevalence of the health conditions for those reporting no disability.

Authors' response: As a data-rich paper, we have sought to keep the numbers presented in the text to a minimum as all corresponding data (prevalence estimates, 95% CIs, AORs, denominators) are shown in the tables, which will appear in the body of the manuscript alongside the text.

Major comments:

Table 1. (pp. 20): consistency needed in the presentation of the unweighted/weighted denominators values in the cells (use "/" or "," in all cells; my preference is "/" to avoid confusions with the "thousands" notation).

Authors' response: We have amended the Tables as per the Reviewer's suggestion.

A description of the full sample (women + men) should be included in the table, or at the very least, characterize how different the two subgroups are, so that we can better interpret the estimates. We would also want to better understand the drivers of the gender differences in outcomes between the disables and non-disabled.

Authors' response: We have described the study's sample in tables 1-4 which give descriptive statistics for all variables considered in the paper, by limiting disability status and stratified by gender. As noted in response to another of Reviewer 6's comments, our paper is already data-rich and so we trust we have provided sufficient detail.

The authors do not present any information regarding the significance of the differences in estimates for women and men. Are these gender differences significant and meaningful? What statistical tests (if any) are used to test these differences?

Authors' response: We have not made comparisons between men and women in this paper given well-established gender differences in sexual behaviour and sexual health outcomes. For this reason, the focus of our paper is on the differences by limiting disability status within gender.

Line 37, pp.20: more clarification needed on the age and education adjustments; what specific model(s) is(are) being estimated? Are the authors interacting age and education with other covariates?

Authors' response: We have expanded our explanation of the variables we adjust for (and why) in the Statistical analysis section of the Methods.

Table 2. pp.22: Does this model adjust for other covariates in Table 1? If not, why? For example, given the significant racial/ethnic differences in disability status (Table 1), one would want to also control for this variable.

Authors' response: Please see our response to Reviewer 5's similar point.

VERSION 2 – REVIEW

REVIEWER	Emmanuel F Drabo Stanford University
REVIEW RETURNED	14-May-2018
GENERAL COMMENTS	My comments to the the earlier draft have been properly handled, hence I have no further comments.